# Taxonomic Re-Evaluation and Genomic Comparison of Novel Extracellular Electron Uptake-Capable *Rhodovulum* *visakhapatnamense* and *Rhodovulum sulfidophilum* Isolates

**DOI:** 10.3390/microorganisms10061235

**Published:** 2022-06-16

**Authors:** Emily J. Davenport, Arpita Bose

**Affiliations:** Department of Biology, Washington University in St. Louis, St. Louis, MO 63130, USA; demily@wustl.edu

**Keywords:** phototrophic bacteria, phototrophic extracellular electron uptake, comparative genomics, transcriptomics, *Rhodovulum sulfidophilum*, *Rhodovulum visakhapatnamense*

## Abstract

*Rhodovulum* spp. are anoxygenic phototrophic purple bacteria with versatile metabolisms, including the ability to obtain electrons from minerals in their environment to drive photosynthesis, a relatively novel process called phototrophic extracellular electron uptake (pEEU). A total of 15 strains of *Rhodovulum sulfidophilum* were isolated from a marine estuary to observe these metabolisms in marine phototrophs. One representative strain, *Rhodovulum sulfidophilum* strain AB26, can perform phototrophic iron oxidation (photoferrotrophy) and couples carbon dioxide fixation to pEEU. Here, we reclassify two *R. sulfidophilum* isolates, strainAB26 and strain AB19, as *Rhodovulum visakhapatnamense* using taxonomic re-evaluation based on 16S and *pufM* phylogenetic analyses. The strain AB26 genome consists of 4,380,746 base-pairs, including two plasmids, and encodes 4296 predicted protein-coding genes. Strain AB26 contains 22 histidine kinases, 20 response regulators, and dedicates ~16% of its genome to transport. Transcriptomic data under aerobic, photoheterotrophy, photoautotrophy, and pEEU reveals how gene expression varies between metabolisms in a novel *R. visakhapatnamense* strain. Genome comparison led by transcriptomic data under pEEU reveals potential pEEU-relevant genes both unique to *R. visakhapatnamense* strains and shared within the *R. sulfidophilum* genomes. With these data we identify potential pEEU-important transcripts and how speciation may affect molecular mechanisms of pEEU in *Rhodovulum* species from the same environment.

## 1. Introduction 

Bacterial metabolisms are diverse, allowing for growth in nearly any environment [1]. Some bacteria are capable of obtaining electrons from insoluble extracellular sources within their environments via a process called extracellular electron uptake (EEU) [2]. Various forms of this metabolism exist with the common theme being that microbes interact with solid-phase conductive substances as sources of electrons [3]. Through EEU, microbes drive biogeochemical cycles in all corners of the earth at undetermined rates [4]. This metabolism also has potential to improve bioremediation of contaminated sites and the biosynthesis of sustainable fuels [5,6,7]. 

EEU was first observed in species such as mineral-reducing, *Shewanella oneidensis* MR1 and the photoautotroph, *Rhodopseudomonas palustris* TIE-1 (hereafter referred to as TIE-1) [8,9,10]. One well-characterized electron uptake system is known; the PioABC operon studied in TIE-1 [11]. Studies of pEEU in TIE-1 revealed the Calvin–Benson–Bassham cycle as an electron sink during pEEU [12,13]. This link to carbon dioxide fixation connects pEEU to the carbon cycle. Furthermore, because TIE-1 can use various soluble and insoluble forms of iron, pEEU connects to the cycling of iron and other elements that are bound to iron minerals [3,14]. Until recently pEEU has been studied solely in freshwater phototrophs, leaving a knowledge gap regarding the prevalence of pEEU in other ecosystems. 

In an effort to address this, we isolated bacteria from a microbial mat in a marine estuary in Woods Hole, Massachusetts, USA (henceforth referred to as Woods Hole isolates) [15,16]. In total, 15 of the Woods Hole isolates were initially identified as *Rhodovulum sulfidophilum*, a purple non-sulfur bacterium capable of many diverse metabolisms including anoxygenic photosynthesis (non-oxygen-evolving). We sequenced and assembled the genomes of 12 of the 15 isolates based on these data (3 isolates, namely *Rhodovulum sulfidophilum* strain AB14, strain AB26, AB30, were assembled previously [15]). We have previously reported that all 15 strains were capable of heterotrophic, photoautotrophic, and photoheterotrophic growth as well as photoferrotrophy [12]. A representative strain, *R. sulfidophilum* AB26, was selected for further growth studies and found to be capable of pEEU in bioelectrochemical systems (BES) using a novel uptake pathway consisting of a di-heme *c*-type cytochrome (EeuP), with carbon dioxide fixation as a cellular sink for acquired electrons [12]. EeuP is found in diverse bacterial lineages without other characterized pEEU uptake systems in their genomes, which may suggest its role as a novel uptake system [12]. 

In assembling and releasing the remaining 12 genomes of the Woods Hole isolates, we discovered 2 isolates, strain AB19 and the representative pEEU strain AB26, that no longer fit into *R. sulfidophilum* but rather the closely related, novel *Rhodovulum visakhapatnamense* species, originally isolated from a tidal seawater enriched under phototrophic conditions [17]. Prior to this study, *R. visakhapatnamense* was represented by a single-strain *R. visakhapatnamense* strain JA181. Like other *Rhodovulum* species, strain JA181 is capable of chemoorganotrophy in the dark and photoheterotrophy under illumination [17]. Currently, three *R. visakhapatnamense* genomes are publicly available: strain AB26, strain AB19, and strain JA181 [17]. Given this new taxonomic classification and the previously described transcriptomic data our group possesses [11], we aim to describe the genome content and expression profiles under these growth conditions to further characterize members of *R. visakhapatnamense*. Comparative genomic analysis based on transcript data of putative pEEU-important transcripts within *R. visakhapatnamense* strain AB26 reveals the distribution of homologous genes within the genomes of the 15 Woods Hole isolates.

## 2. Materials and Methods

### 2.1. Microbial Isolation and Cultivtion Conditions

As previously described [11], Rhodovulum sulfidophilum strains were isolated in July 2014 from independent microbial mat samples from the Trunk River estuary in Woods Hole, MA. Enrichments were cultivated photoheterotrophically in anoxic artificial seawater (SW) medium supplemented with 20 mM acetate or 10 mM sodium thiosulfate. Enrichments were cultivated with ∼850-nm light at 30 °C, and passaged six times, followed by streaking aerobically six times on Bacto agar with Difco marine broth (MB) 2216 (BD Diagnostic Systems, Hunt Valley, MD USA) to isolate single colonies. All growth experiments were carried out at 30 °C unless otherwise noted. AB26 strains were grown in MB broth under dark with shaking at 200 rpm for aerobic heterotrophic growth. All phototrophic (photoautotrophic and photoheterotrophic) growth experiments were carried out anaerobically in SW media supplemented with 70 mM sodium bicarbonate and 1 mM sodium sulfate in sealed sterile serum bottles. The phototrophic cultures were grown without shaking under light (with a single 60 W incandescent light bulb at 25 cm). For anaerobic photoautotrophic growth, AB26 strains were grown with H_2_ (80% H_2_: 20% CO_2_ at 50 kPa) or 10 mM sodium thiosulfate or 5 mM Fe(II) chloride in SW medium. To prevent iron precipitation, the photoautotrophic culture with Fe(II) contained 10 mM nitrilotriacetic acid (NTA). For anaerobic photoheterotrophic growth, AB26 strains were grown in SW medium with 10 mM acetate. The headspace atmosphere of the phototrophic cultures consisted of 80% N_2_ and 20% CO_2_, except for hydrogen. Where a change in culture medium was required, cells were washed three times in basal SW medium post-centrifugation at 5000× *g*.

### 2.2. Genome Sequencing

Genome analysis was performed as previously described [15]. Genomic DNA was isolated from mid-log phase cell culture in marine broth using the DNeasy Blood and Tissue Kit (Qiagen, Germantown, MD, USA). Samples were prepped for Illumina 250-bp paired-end sequencing using Nextera sample prep kit (Illumina, Inc., San Diego, CA, USA), and sequenced on Illumina MiSeq platform with V2 chemistry. Reads were quality checked and adapter trimmed using Trimmomatic version 0.33 using default parameters for paired-end reads [18]. Reads were de novo assembled using CLC Genomics Workbench (CLC Bio-Qiagen, Aarhaus, Denmark), and scaffolds generated using MeDuSa [19] and *R. sulfidophilum* DSM 2351 as an alignment guide. Alignment of reads to *R. sulfidophilum* DSM 2351 was performed using Bowtie2 version 2.2.29 [20] short read mapper, and annotated with National Center for Biotechnology Information Prokaryotic Genome Annotation Pipeline [21].

### 2.3. Phylogenetic Analysis, Taxonomic Re-Evaluation, and Comparative Genomics

ANI between strains was performed using JSpeciesWS [22] and visualized with Morpheus (Morpheus, https://software.broadinstitute.org/morpheus (accessed on 3 February 2022)). Phylogenetic trees were built using MEGA11 [23]. For strain-level phylogenetic analysis, 16S rRNA sequences and the photosynthetic reaction center subunit M (*pufM*) protein sequences were used to build trees [24,25]. The 16S tree used Kimura 2-parameter model [26] and a gamma distribution of 5. The *pufM* tree used Jones-Taylor-Thornton (JTT) model [27] and a gamma distribution of 5. For phylogenetic analysis of EeuP protein sequences among *R. sulfidophilum* strains, the JTT model was used with a gamma distribution of 5. All sequences were aligned using the ClustalW algorithm. BLASTp was performed using a local protein database (Appendix A).

### 2.4. RNA Isolation, RNA Sequencing, and Differential Expression Analysis

As previously described [12], Cell cultures were sampled anaerobically and immediately mixed with 1:1 with RNAlater^®^ (Qiagen, Germantown, MD, USA). RNA was extracted from cells using the RNeasy Mini Kit (Qiagen, USA) and DNA removal performed using Turbo DNA-*free* Kit (Ambion, Austin, TX, USA). RNA samples were tested for purity using PCR. Illumina single-end 50-bp libraries were prepared and sequenced at Washington University’s Genome Technology Access Center on an Illumina HiSeq3000 (Illumina Inc., Madison, WI, USA). Reads were mapped to the strain AB26 genome using TopHat2 version 2.1.1 and the gff3 annotation file as a guide for sequence alignment. Bowtie 2 version 2.3.3.1 was used to index the reference genome FASTA file. The number of reads mapping to each feature were counted by HTSeq version 0.9.1. Differentially expressed genes were determined in DESEQ2 version 1.16.1 using the HTSeq read counts. Significantly differentially expressed genes were identified using an adjusted *p*-value cutoff of 0.05. 

### 2.5. Supplemental Materials & Methods

Appendix A can be found in the Appendix A. Accession numbers: *Rhodovulum visakhapatnamense* strains AB26 GCA_001941715.1; AB19 GCA_016757185.1; JA181 GCA_004365965.1; *Rhodovulum sulfidophilum* strains: AB14 GCA_001941695.1; AB15 GCA_021409265.1; AB16 GCA_016757145.1; AB17 GCA_021409205.1; AB18 GCA_021409185.1; AB20 GCA_016757195.1; AB21 GCA_021409225.1; AB22 GCA_016757115.1; AB23 GCA_016757055.1; AB28 GCA_021409165.1; AB30 GCA_001941745.1; AB33 GCA_021568715.1; AB35 GCA_016757045.1; DSM1374 GCA_001633165.1; DSM2351 GCA_001548075.1; IM796 GCA_016653205.1; SE1 GCA_010119435.1; SNK001 GCA_001633145.1.

## 3. Results and Discussion

### 3.1. Taxonomic Re-Evaluation of Strain AB26 and Strain AB19

A total of 15 isolates were cultured from a marine microbial mat in Woods Hole, MA, USA and identified as belonging to *R. sulfidophilum* through 16S sequencing [12,15]. Prior to sequencing the Woods Hole isolates, few *R. sulfidophilum* genomes were publicly available. Whole genome alignments of strain AB26 showed only 57.8% of the genome sequence aligned to the available reference genome *R. sulfidophilum* DSM2351 [15]. Growth experiments of select isolates revealed differences in doubling times in the same growth conditions [12], and therefore we sought to further investigate the phylogenetic relatedness of the isolated strains. 16S and photosynthetic reaction center subunit M (*pufM*) phylogenetic trees were produced to determine the relatedness between isolates from Woods Hole (Figure 1).

16S phylogenetic analysis shows strain AB26 and *R. sulfidophilum* strain AB19 reside in a separate branch from all other Woods Hole isolates, with a second clade consisting of strain AB33, strain AB16, and strain AB30. The remaining Woods Hole isolates generally nest within a group together. The pattern of the 16S phylogenetic tree is mirrored in the photosynthetic reaction center subunit M (*pufM*) tree in that strain AB26 and strain AB19 again reside in a separate branch from the remaining Woods Hole isolates. Further investigation using average nucleotide identities (ANI) revealed strain AB26 and strain AB19 ANI values are far below the cut-off value for same-species relatedness (>~94%, Figure 2) [27]. The remaining Woods Hole isolates all show ANI values indicative of same-species relatedness. Two groups of strains, AB18 and AB33 and strains AB35 and AB15, show highest similarity to each other. One other isolate, strain AB28, showed lower ANI values to all other strains (94–95% ANI) but were still within same species-values (Figure 2).

To determine the proper species designation of strain AB26 and strain AB19, ANI comparisons between strain AB26 and strain AB28 were performed to include the closely related species *Rhodovulum viride* and *Rhodovulum visakhapatnamese* (Appendix A) [17,28]. Strain AB28 did not show any increased similarities to other *Rhodovulum* genomes, suggesting the current taxonomy is correct, but may change as more *Rhodovulum* genomes are assembled. ANI analysis shows strain AB26 and strain AB19 share ANI values of 97.95% with *R. visakhapatnamense*, placing strain AB26 and strain AB19 in a new species (Figure 3).

### 3.2. Strain AB26 Genome Features and Expression Analysis

Strain AB26 was chosen as a representative strain for this study due to its preferable growth characteristics [12]. The genome of strain AB26 consists of 4,380,746 bp of DNA and a GC content of 67.9%, assembled into 3 scaffolds. strain AB26 has 4375 genes, with 4296 protein-encoding genes (Table 1). The strain AB26 genome contains genes allowing for diverse metabolic capabilities that purple non-sulfur bacteria are known for. This is in line with previous observations that strain AB26 is capable of aerobic heterotrophic growth in the dark, photoheterotrophy using succinate and acetate for carbon and electrons, and photoautotrophy, using inorganic substances, including thiosulfate and hydrogen (H_2_).

#### 3.2.1. Phototrophy

*R. visakhapatnamense* is a member of the purple nonsulfur bacteria capable of anoxygenic photosynthesis [29]. The genes necessary for growth via photosynthesis occur in a ~50-kb region of the genome spanning four separate operons. The first operon spans a 13-kb region and encodes the reaction center, bacteriochlorophyll synthesis, carotenoid synthesis, and light-harvesting proteins. Reaction center subunits M (BV509_00330) and L (BV509_00335) share 70% identity with the reaction center genes in the model anoxygenic phototroph *Cereibacter sphaeroides* [30]. The second photosynthetic gene cluster (PGC) consists of tetrapyrrole biosynthesis genes *bchEJ*. The third PGC is homologous to the *puc* operon, which encodes the structural polypeptides of the light-harvesting-II peripheral antenna complex, and also contains a PucC family protein [29]. PucC encoded in model *puc* operons regulates response to oxygen tension and light intensity [30]. The fourth PGC spans 32-kb and contains further pigment synthesis genes, the photosynthetic reaction center subunit H, and various response regulators.

Transcriptomic data shows photosynthetic reaction center subunit M is significantly expressed during pEEU compared to aerobic growth (log_2_fold change ~1; *p*  <  0.0001), but not when grown with H_2_ compared to aerobic growth (log_2_fold change ~0.5; *p* > 0.05). The same is true for photosynthetic reaction center subunit L during pEEU compared to aerobic (log_2_fold change ~1; *p*  <  0.05), but not growth with H_2_ compared to aerobic growth (log_2_fold change ~0.5; *p* > 0.05).

#### 3.2.2. Carbon Dioxide Fixation

The strain AB26 genome contains form I (BV509_05525) and form II (BV509_05520) of RuBisCO, the enzyme necessary for carbon dioxide fixation via the Calvin–Benson–Bassham (CBB) cycle [29]. The two forms reside in an operon together with the CbbQ RuBisCO-activating protein (BV509_05515), and a LsyR family transcriptional regulator (BV509_15210), which shares similarity with the RuBisCO regulator CbbR. The genes required for photorespiration via the Calvin–Benson–Bassham cycle are housed 2-Mb upstream from the RuBisCO enzyme gene cluster. As previously described [12], transcriptomic data reveals upregulation of form I RuBisCO (log_2_fold change ~2, *p* < 0.05) but downregulation of form II RuBisCO (log_2_fold change ~−0.35, *p* > 0.1) during pEEU, and upregulation of a formate dehydrogenase operon (log_2_fold change ~4; *p*  <  0.0001) [12]. The high upregulation of formate dehydrogenase may suggest an involvement of this enzyme in catalyzing CO_2_ fixation [31].

#### 3.2.3. Reducing Power

Strain AB26 is metabolically diverse, capable of aerobic heterotrophy, anaerobic photoheterotrophy, or anaerobic photoautotrophy using various inorganic compounds for electron donors (Figure 4). Strain AB26 contains a 16-kb operon coding for the synthesis and assembly of a nickel-dependent hydrogenase (BV509_06105-BV509_06195, Figure 4a). Expression of this nickel-dependent hydrogenase was highest during growth with thiosulfate compared to aerobic growth, with expression of genes within the operon ranging from ~1–3.5-fold (*p* < 0.001, Figure 4a). Strain AB26 is also capable of growth using thiosulfate via *sox*-like genes organized into 3 clusters. The first cluster contains two SoxXY-like carrier proteins (BV509_04530- BV509_04535) and a single metallohydrolase SoxH (BV509_04575). The second cluster contains the canonical *soxABCXYZ* genes (BV509_09605-BV509_09655). *soxR* (BV509_15000) is 1-Mb downstream, and is a putative transcriptional regulator of the *soxABCXYZ* genes [32]. *soxR* is slightly upregulated during growth with thiosulfate compared to aerobic (log_2_fold change ~1.3; *p*  <  0.0001, Figure 4b), which corresponds with the increased expression of other genes in the *sox* operon. However, *soxR* is not upregulated during growth with acetate as compared to aerobic growth (log_2_fold change ~−0.3; *p*  <  0.0001), despite increased expression patterns of the other *sox* genes (1–2-fold; *p*  <  0.0001, Figure 4b). A carbon monoxide dehydrogenase (BV509_12715-BV509_12730, Figure 4a) and a formate dehydrogenase operon (BV509_15145-BV509_15165, Figure 4a) are also present in the strain AB26 genome.

#### 3.2.4. Regulation and Signaling

To be successful in transient environments requires genomic resources to sense and respond. Strain AB26 dedicates ~5% of its genome to signal transduction and regulation. Typical bacterial genomes dedicate 5–6% to regulation and sensing. Some purple non-sulfur bacteria such as *R. palustris* TIE-1 allot ~10% of their genome to environmental sensing, similar to other soil bacteria [34]. Strain AB26 contains 22 histidine kinases, 4 of which contain an HAMP domain (domain present in Histidine kinases, Adenylyl cyclases, Methyl-accepting proteins and Phosphatases) involved with chemotaxis [35]. Thirteen PAS domain-containing proteins are likely involved in light and oxygen sensing [36], and 2 GAF domains are potentially associated with photoreceptors [37]. Strain AB26 contains 19 RNA polymerase sigma factors and 121 transcriptional regulators belonging to 30 different families. During pEEU in strain AB26, transcripts encoding histidine kinases responding to envelope stress (BV509_04670, log_2_fold change ~2, *p* < 0.0001, Figure 5a), nitrate assimilation (BV509_11520, log_2_fold change ~3, *p* < 0.0001, Figure 5b), a sigma-70 factor involved in exocytoplasmic stress (BV509_00135, log_2_fold change ~2, *p* < 0.01, Figure 5e),and an anti-sigma antagonist (BV509_19770 log_2_fold change ~3, *p* < 0.0001, Figure 5d) were upregulated. This suggests that strain AB26 is potentially under stress during pEEU compared to growth under H_2_. Also upregulated during pEEU is a flagellar switching response regulator (BV509_14575, (log_2_fold change ~2, *p* < 0.0001, Figure 5c).

#### 3.2.5. Biodegradation and Carbon Storage

The diverse metabolisms of purple non-sulfur phototrophs allow for proliferation in industrial waste environments [38]. *Rhodopseudomonas palustris* and *Cereibacter sphaeroides* are two purple phototrophs known for their ability to metabolize diverse compounds including fatty acids, amino acids, aromatic compounds, and lignin. They are, therefore, important species in the bioremediation of wastewater and polluted environments [39,40,41,42]. Some purple phototrophs are also capable of synthesizing carbon storage polymers such as polyhydroxalkanoates (PHA), which they degrade in carbon-starved environments. The genome of strain AB26 encodes a variety of genes for the metabolism of aromatic compounds: a benzoate transporter (BV509_16875) and a 4-hydroxybenzoate octaprenyltransferase (BV509_17190) for the first step of benzoate degradation, a salicylate esterase (BV509_RS17105), a biphenyl-2,3-diol 1,2-dioxygenase III-related protein (BV509_11920) for the degradation of biphenyl, a type II 3-dehydroquinate dehydratase (BV509_13850) for the degradation of quinate, and for the degradation of gentisate, a fumarylacetoacetate hydrolase family protein (BV509_03940) and a maleylacetoacetate isomerase (BV509_16060). Strain AB26 contains two operons for the degradation of phenylacetate (BV509_03065-BV509_03095, BV509_05275-BV509_05285) [43]. Similar to *R. palustris* TIE-1 [42], strain AB26 encodes genes for the synthesis and catabolism of polyhydroxybutyrate (PHB). *phaA*, which encodes a beta-ketothiolase, and *phaB*, which encodes an acetoacetyl-CoA reductase, are organized separately from each other and the remaining PHB gene cluster. The remainder of the PHB cluster contains the PHA depolymerase gene *phaZ*, the class 1 poly(R)-hydroxyalkanoic acid synthase *phaC*, which encodes the phasin protein affecting PHB granule accumulation and utilization, and the PHA synthesis repressor gene *phaR* (BV509_06270-BV509_06285). Unlike TIE-1, strain AB26 only has single-gene copies for the synthesis and utilization of PHB. As previously described [12], the PHB biosynthesis enzymes were downregulated during pEEU as compared to aerobic growth (log_2_fold change ~0, *p* > 0.05), and the repressor *phaR* was slightly upregulated (log_2_fold change ~1, *p* < 0.01).

#### 3.2.6. Transporters

The strain AB26 genome encodes 561 transport system genes, dedicating ~16% of its genome to transport. Of these, 347 transport genes are involved in primary transport, with 45 different ABC transport systems and 23 ATPase genes. Strain AB26 contains type III secretion system genes for flagella biosynthesis, and seven type II secretion system families (Figure 6).

In total, 60 of the ABC transport genes were identified as various amino acid transport systems, with 22 genes dedicated to the transport of branched chain amino acids. For iron acquisition, strain AB26 contains 11 *tonB* genes and 5 iron transport systems. The lack of siderophore biosynthesis genes suggests that strain AB26 may transport siderophores produced by other bacteria into the cell. Strain AB26 contains two ferrous iron transport genes (BV509_02645-BV509_02650) for aerobic iron acquisition. Nutrient acquisition (Figure 6) for strain AB26 includes urea transport (BV509_09920-BV509_09945) and sulfate transport (BV509_13630-13645). Located on a plasmid are various metal uptake systems [15] including zinc (BV509_08850, BV509_19470, BV509_19480, BV509_20850), nickel (BV509_20695-BV509_20715, BV509_11055), and manganese (BV509_03550, BV509_20820-20825, Figure 6), which may help maintain metal ion homeostasis. Strain AB26 contains four microcin transporters, 14 multidrug transporters, and 9 efflux transporters. The Sec pathway was upregulated compared to aerobic growth only during pEEU, supporting the requirement of protein export in the periplasmic space to facilitate extracellular electron transfer across the membranes (Figure 6)

Strain AB26 contains 176 passive transport genes, with two multidrug and toxic compound extrusion (MATE) genes, 10 resistance-nodulation-cell division (RND) pumps, 23 major facilitator superfamily (MFS) genes, and 59 tripartite ATP-independent periplasmic (TRAP) transport genes. The TRAP family transporters are common in marine bacteria such as strain AB26 due to the natural Na^+^ gradient in their environment [44].

#### 3.2.7. Nitrogen Assimilation and Fixation

Strain AB26 uses the glutamine synthetase and glutamate synthase for ammonia assimilation in the cell. Strain AB26 genome contains two ammonia transporters, three glutamine synthetases, and a glutamine–synthetase adenyltransferase to regulate activity of glutamine–synthetases. Strain AB26 also encodes genes to convert urea to ammonia (BV509_09920-BV50-0995, Figure 7). Strain AB26 genome contains structural genes for a single molybdenum nitrogenase (BV509_14955-BV509_14965), assembly and cofactor genes, and the nitrogen fixation-specific sigma factor RpoN to fix nitrogen during anaerobic growth (Figure 7). During pEEU, one glutamine synthetase is highly upregulated (BV509_16830, log_2_fold change ~2.5, *p* < 0.0001), a second glutamine synthetase is slightly upregulated (BV509_16820, log_2_fold change ~0.75, *p* < 0.001; Figure 7a), and both are downregulated during photoheterotrophic growth compared to aerobic growth. Conversely, during pEEU, the glutamate synthase is downregulated as compared to aerobic growth (Figure 7a).

#### 3.2.8. Extracellular Electron Uptake (EEU) Genes

To perform pEEU, Gram-negative bacteria require uptake systems consisting of electron transfer proteins spanning the outer membrane, periplasmic space, and inner membrane. These systems may also include porin proteins to span membranes and facilitate the exchange of electrons from the cell exterior to the interior. While some characterized electron uptake systems exist [11,45], these uptake systems or homologs of them are not present in all bacteria that are hypothesized to be able to perform pEEU. Our group recently identified a novel protein important to phototrophic extracellular electron uptake in strain AB26, which does not encode any known EEU pathways in its genome; the di-heme *c*-type cytochrome EeuP [12]. As previously reported, EeuP is a novel pEEU-important protein found in A26 but also in species spanning the Proteobacteria and Acidobacteria. So far, no other components of the novel EEU pathway have been identified in strain AB26 other than EeuP. To identify other potential proteins that might be important in pEEU, a RNASeq guided comparative genomic approach with specific targets, such as cytochrome c-containing proteins, would be useful. This is because these proteins are often associated with extracellular electron transfer [3].

### 3.3. RNA-Seq Led Genomic Comparison between Rhodovulum *spp.*

Our group previously performed RNA-Seq in strain AB26 between photoautotrophy with H_2_/CO_2_ as electron donor/carbon source compared to pEEU via poised electrodes [12] in an effort to characterize the molecular mechanisms of pEEU in this novel isolate and identify the cellular sinks of electrons derived from extracellular sources. With these data, and the full genome sequences of the 14 other isolates comprising two *Rhodovulum* species, we can identify homologs of pEEU-important transcripts and infer ability to perform pEEU based on their presence or absence in each strain. Appendix A shows the general genome features of all 15 Woods Hole isolates and publicly available genomes for both *R. visakhapatnamense* [46] and *R. sulfidophilum* DSM 2351 [47] and DSM1374 [48].

#### 3.3.1. C-Type Cytochromes (Cyt_c_)

Electron transfer with solid phase donors is commonly carried out by *c-*type cytochromes. *c*-type cytochromes mediate single electron transfers via a covalently bonded heme group [49,50], making them ideal candidates for pEEU processes. Therefore, our analysis of the pEEU vs. hydrogen transcript data focuses on identifying upregulated transcripts encoding *c*-type cytochromes, or upregulated hypothetical proteins and searching for the canonical CXXCH (where X is any amino acid residue) heme-binding motif within the protein sequence.

##### Diheme Cyt_c_ EeuP

RNA-Seq identified a significantly upregulated transcript (log_2_fold change ~2, *p* < 0.0001) in strain AB26 encoding the hypothetical protein later identified to be the di-heme EeuP (BV509_10070). BLASTp analysis identified homologous sequences of EeuP in all isolates except strain AB21. BLASTp results for strain AB21 returned two partial cyt*_c_* sequences and one partial hypothetical protein sequence. These partial sequences separately align with the intact EeuP homologs and may be an artifact of assembly. From this we can hypothesize that all the isolates are pEEU-capable. Maximum likelihood phylogenetic analysis of the EeuP protein sequence (Figure 8) reveals EeuP in the isolates reflects the same relationship shown in the 16S and *pufM* phylogenetic trees (Figure 1), in that strain AB19 and strain AB26 EeuP sequences group outside of the *R. sulfidophilum* EeuP sequences. As previously reported [12], EeuP homologous sequences were also identified in Alpha-, Beta-, and Gammaproteobacteria classes and in one *Acidobacter* species. Further analysis of the species containing EeuP homologs may reveal the relationship between EeuP conservation and metabolic/environmental adaptation.

##### Hypothetical Protein BV509_18570

Another significantly upregulated transcript (log_2_fold change ~6, *p* < 0.0001) in strain AB26 encodes a hypothetical protein (BV509_18570). BV509_18570 is a 413-residue long protein containing a single heme-binding motif (CXXCH). InterProScan [51] predicts the molecular functions as electron transfer and heme binding. PRED-TAT [52] did not predict a secretory signal peptide (Sec), which would be expected for an electron transport protein involved in trans-membrane transport. Surprisingly, BLASTp analysis revealed no homologs of this open reading frame (ORF) within any of the *R. sulfidophilum* isolates. Homologs of BV509_18570 were found in *R. visakhapatnamense* strain AB19 (JMJ92_02255) and *R. visakhapatnamense* strain JA181 (EV657_1244). The homologous sequences are each 625 amino acid residues in length, ~200 amino acid residues longer than the strain AB26 BV509_18570. Protein sequence alignments in MAUVE [53] show JMJ92_02255 and EV657_1244 sequences align 212 residues before aligning to BV509_18570, and these additional residues contain a second heme-binding motif and a Sec signal peptide sequence. Therefore, it is possible the 413-residue long BV509_18570 may be truncated due to an assembly error and, similar to the related homologs, contains an additional heme-binding motif and Sec signal peptide sequence. BV509_18570 could be a another di-heme cyt_c_ similar to EeuP, which was also annotated as a hypothetical protein, and be important for pEEU specific to *R. visakhapatnamense* spp. This hypothesis is supported by the high log_2_fold change of this transcript during pEEU and presence of a Sec signal peptide sequence, suggesting the transport of this protein across the inner membrane to the periplasmic space.

##### Diheme Cyt_c_ BV509_14915

This transcript (log_2_fold change ~6, *p* < 0.0001) encodes a cyt*_c_* with two heme-binding motifs. BV509_14915 is 347 residues long and contains a Sec signal peptide. Because of this, BLASTp analysis shows all 15 marine *Rhodovulum* spp. isolates contain a homolog of this di-heme cyt*_c_*. Given the high log_2_fold change, Sec signal peptide sequence, heme-binding motifs, and the presence of homologs within the other isolates’ genomes, this cytochrome could be a conserved component of the pEEU molecular pathway in the *Rhodovulum* spp. isolates.

##### Cyt_c_ BV509_15885

The transcript encoding the gene BV509_15885 is annotated as a cyt*_c_*. RNA-Seq data did not identify this cyt*_c_* as significantly upregulated during pEEU (log_2_fold change −0.48), but it was identified through heme staining (membrane fraction) followed by mass spectroscopy compared to hydrogen grown cultures [12]. This cyt*_c_*contains 5 heme-binding motifs and a predicted Sec signal peptide. BLASTp analysis identified homologs of this cyt*_c_*in strain AB19 and *R. visakhapatnamense* strain JA181. Homologs in the *R. sulfidophilum* isolates show 71–75% identity.

#### 3.3.2. Transcriptional Regulators

To understand expression patterns and potential regulators of pEEU genes, the pEEU vs. hydrogen transcript data was analyzed for any upregulated transcriptional regulators.

##### Mar Family Transcription Regulator BV509_01095

BV509_01095 encodes a hypothetical protein identified as a member of the multi antibiotic resistance family transcriptional regulator (log_2_fold change ~6, *p* < 0.001). InterProScan predicts phosphorelay signal transduction function. BV509_01095 is present only in strain AB19 and strain AB26 via BLASTp analysis, but not *R. visakhapatnamense* strain JA181 or any of the *R. sulfidophilum* isolates.

##### Transcriptional Regulator BV509_15825

This transcript is another transcriptional regulator upregulated during pEEU in strain AB26 (log_2_fold change ~4, *p* < 0.0001). This regulator is also predicted to be involved in signal transduction response regulation via InterProScan, with DNA-binding wing-helix domain and tetratricopeptide repeat domain. BLASTp reveals this transcription regulator is unique to strain AB19, strain AB26, and *R. visakhapatnamense* strain JA181 but not the *R. sulfidophilum* isolates.

## 4. Conclusions

With the current accessibility of genome sequencing and ability to isolate bacterial species from diverse environments, this paper shows that taxonomy requires continuous addressing. Here, we provide a current taxonomic classification of 15 *Rhodovulum* spp. isolated from a marine estuary, specifically the re-classification of two of the original 15 isolates previously identified as *R. sulfidophilum* now as *R. visakhapatnamense*. These isolates are capable of many modes of metabolism, including photoferrotrophy and pEEU. To expand the characterization of *R. visakhapatnamense*, which at present is represented by 3 genomes assembled at the scaffold level, we describe the genome contents of *R. visakhapatnamense* strain AB26, and pair the genome data with expression data under anaerobic photoheterotrophy (acetate), photoautotrophy (H_2_ or thiosulfate as electron donor), and photosynthetic extracellular electron uptake (poised electrode as electron donor). Upregulated transcripts under pEEU of interest to the analysis include *c*-type cytochromes and transcriptional regulators. BLASTp shows some transcripts of interest are unique to *R. visakhapatnamense* genomes and could be indicative of adaptation of molecular mechanisms for pEEU explained by speciation.

What will be interesting to observe in the future is continued comparison of expression profiles and proteomes under pEEU and photoferrotrophy between *R. sulfidophilum* and *R. visakhapatnamense* strains from Woods Hole, as this data points to differences between species. Mutational analysis of genes of interest in each species will be necessary to determine their role under these growth conditions. With the data presented here, we can begin to characterize the metabolic pathway of pEEU in *Rhodovolum* species.

## Figures and Tables

**Figure 1 microorganisms-10-01235-f001:**
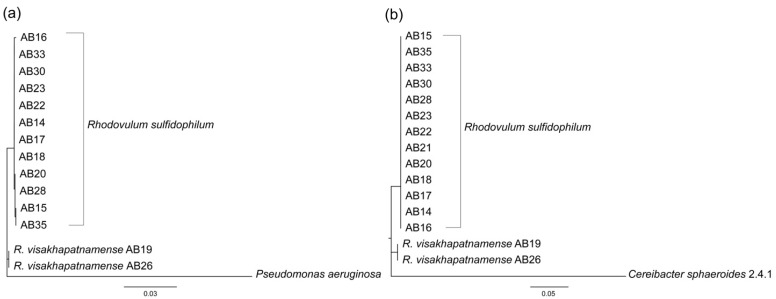
16S rRNA (**a**) and *pufM* (**b**) phylogenetic trees between Woods Hole *R. sulfidophilum* isolates. (**a**) 16S phylogenetic tree constructed by maximum-likelihood method using the Kimura-2 parameter model [26], tested by bootstrapping (500 re-samplings) in MEGA11 [22] with *Pseudomonas aeruginosa* as outgroup. (**b**) *pufM* tree constructed by maximum-likelihood using the Jones–Taylor–Thornton parameter model [27], tested by bootstrapping (500 re-samplings) in MEGA11 [22], using *Cereibacter sphaeroides* 2.4.1 as outgroup.

**Figure 2 microorganisms-10-01235-f002:**
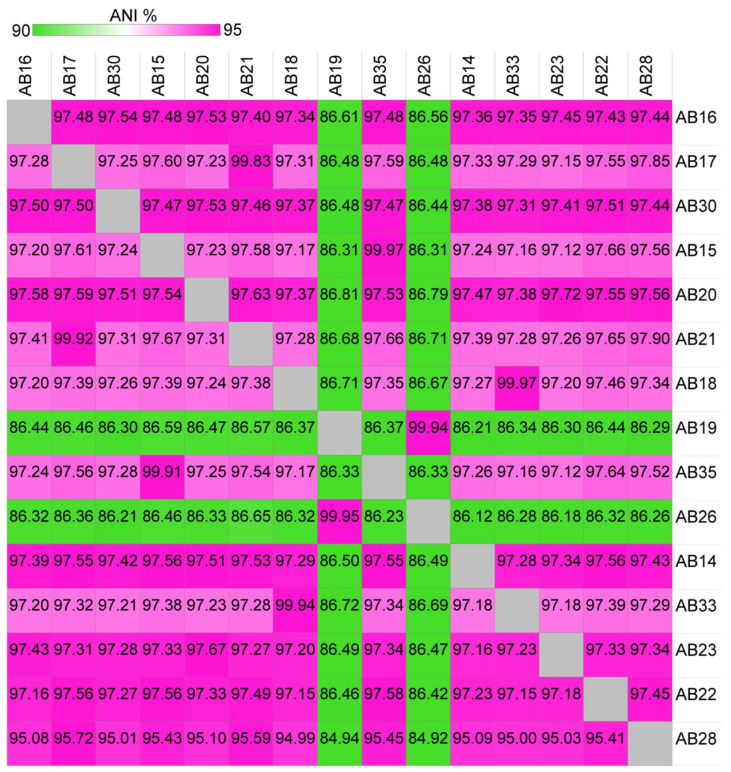
Heatmap showing average nucleotide identities (ANI) values from pairwise comparisons of the 15 Woods Hole isolates. ANI values were calculated using JSpeciesWSd [22] and visualized as a heatmap using Morpheus (Morpheus, https://software.broadinstitute.org/morpheus (accessed on 3 February 2022).

**Figure 3 microorganisms-10-01235-f003:**
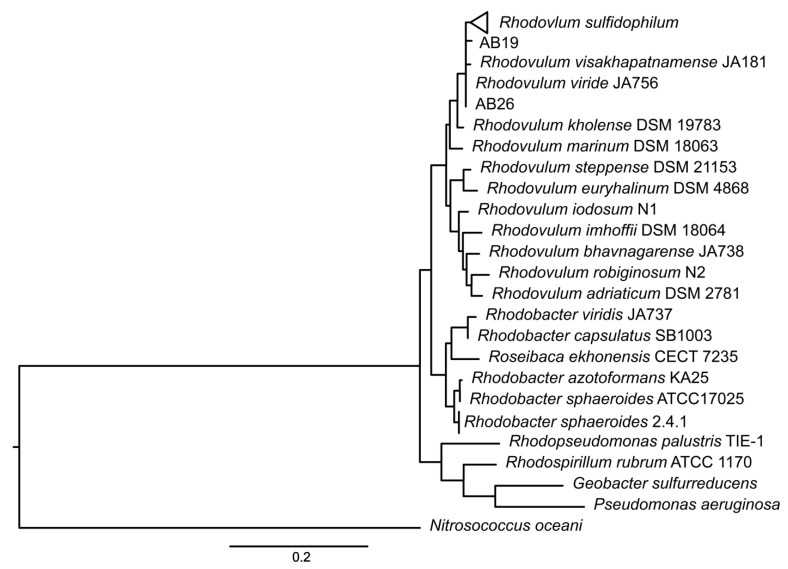
16S phylogenetic tree showing the new taxonomic assignment of strain AB19 and strain AB26 as *Rhodovulum visakhapatnamense*. Tree constructed by maximum-likelihood method using the Kimura-2 parameter model [26], tested by bootstrapping (500 resamplings) in MEGA11 [23].

**Figure 4 microorganisms-10-01235-f004:**
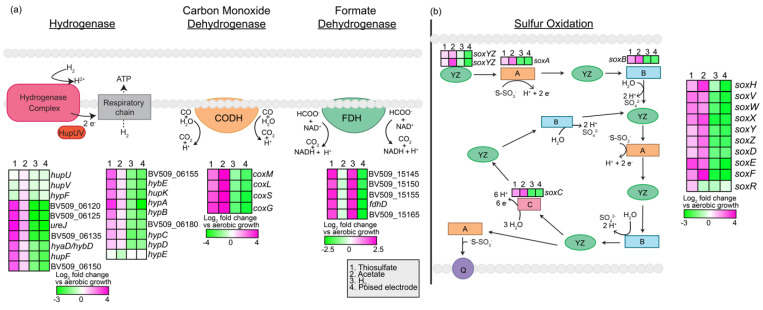
Genes involved in oxidation of inorganic compounds as a source of reducing power and their expression analysis via RNASeq. (**a**) nickel-dependent hydrogenase, carbon monoxide dehydrogenase, and formate dehydrogenase and their expression based on RNASeq. (**b**) sulfur oxidation pathway [33] and expression of the putative genes based on RNASeq. Heatmaps describe expression of four modes of metabolism (thiosulfate, acetate, H_2_, and pEEU) compared to aerobic growth. (**a**) CODH (carbon monoxide dehydrogenase), FDH (formate dehydrogenase), CO (carbon monoxide), HCOO^−^ (formate) (**b**) Q (quinone pool).

**Figure 5 microorganisms-10-01235-f005:**
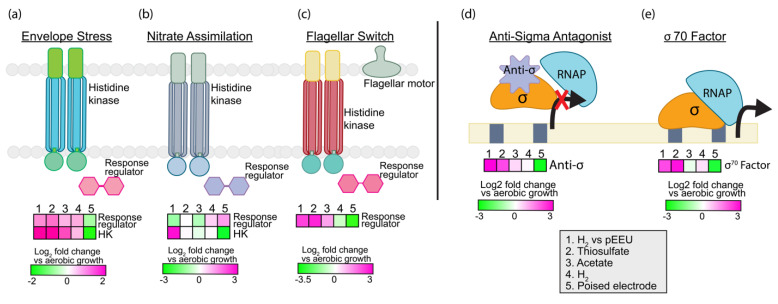
Expression analysis of significantly upregulated environmental responses such as envelope stress (**a**) nitrate assimilation (**b**) and flagellar switch (**c**) under pEEU compared to photoautotrophy with H_2_ (growth condition 1). Heatmaps also describe expression of transcription regulators anti-sigma antagonist (**d**) and sigma 70 factor (**e**) under four modes of metabolism (thiosulfate, acetate, H_2_, and pEEU) compared to aerobic growth.

**Figure 6 microorganisms-10-01235-f006:**
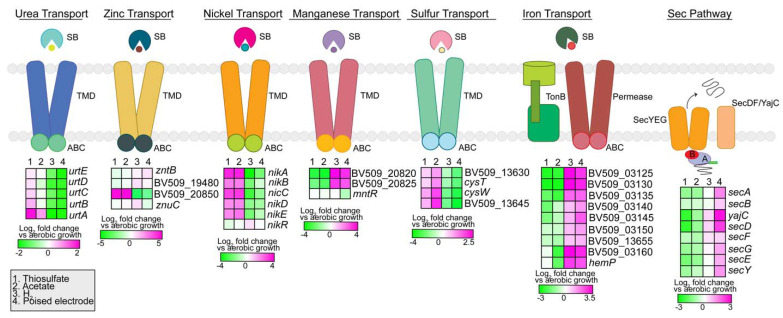
Expression analysis of transport systems in strain AB26. The Sec pathway genes are all upregulated during pEEU, which supports the need for protein export into the periplasmic space to facilitate extracellular electron transport. Heatmaps describe expression of four modes of metabolism (thiosulfate, acetate, H_2_, and pEEU) compared to aerobic growth. SB (substrate binding domain), TMD (transmembrane domain), ABC (ATP cassette binding domain).

**Figure 7 microorganisms-10-01235-f007:**
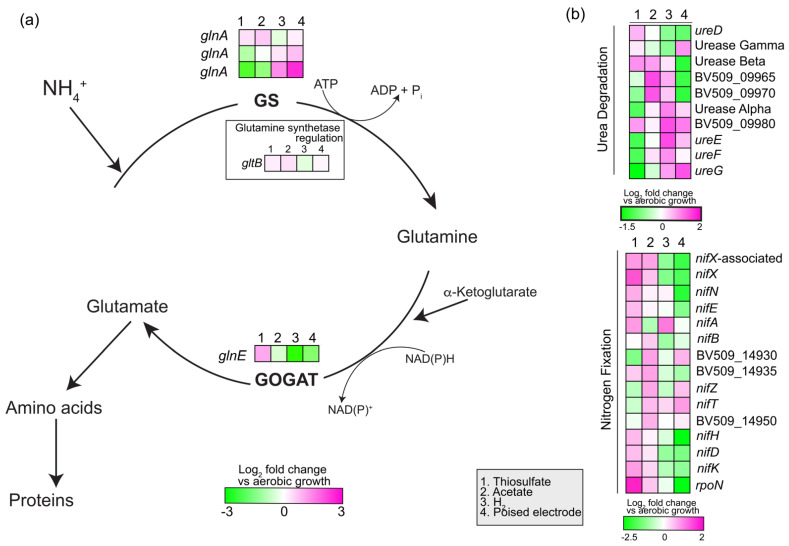
Expression analysis of genes involved in the glutamine synthetase/glutamate synthase (GS/GOGAT) pathway for ammonia assimilation in strain AB26 (**a**). Gene expression in urea degradation and nitrogen fixation (**b**). Heatmaps describe expression of four modes of metabolism (thiosulfate, acetate, H_2_, and pEEU) compared to aerobic growth. (**a**) GS (glutamine synthetase), GOGAT (glutamate synthase) (**b**) BV509_09965 (hypothetical protein), BV509_09970 (DUF1127 domain-containing protein), BV509_09980 (DUF1127 domain-containing protein), BV509_14930 (4Fe-4S dicluster domain-containing protein), BV509_14935 (hypothetical protein), BV509_14950 (SIR2 family protein).

**Figure 8 microorganisms-10-01235-f008:**
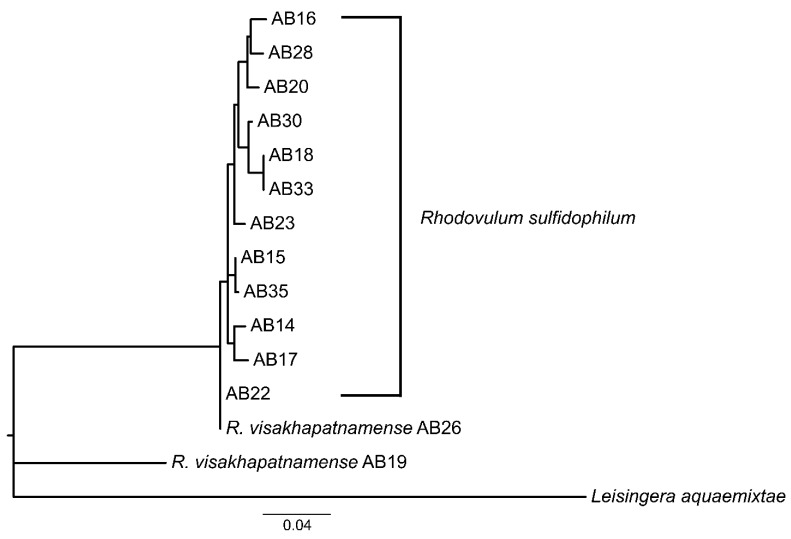
Phylogenetic tree showing protein sequence similarity of the diheme EeuP in Woods Hole isolates. Tree constructed by maximum-likelihood method using the Kimura-2 [25] parameter model, tested by bootstrapping (500 resamplings) in MEGA11 [22].

**Table 1 microorganisms-10-01235-t001:** Genome features of strain AB26. Data for table derived from NCBI and JGI IMG. Strain AB26 is also capable of photosynthesis via pEEU, using electrons obtained from a poised electrode (pEEU) or photoferrotrophy, obtaining electrons by oxidizing Fe(II) [12].

Genome Size (bp)	4,380,746	
**GC%**	67.9	
**Scaffolds**	3	
**Genes total number**	4375	**% Total**
Protein coding genes with predicted function	3212	73.42
rRNA genes	3	
tRNA genes	49	
Other RNA genes	7	
Protein coding genes with function prediction	3212	73.4
Without function prediction	1084	24.8
Biosynthetic gene clusters	5	
Genes in biosynthetic clusters	115	2.6
Protein coding genes coding signal peptides	370	8.5
Protein coding genes coding transmembrane proteins	919	21.0
COG clusters	1869	56.4
Pfam clusters	2235	65.0
TIGRfam clusters	1083	83.6

## Data Availability

Sequencing reads were deposited in the NCBI database under BioProject PRJNA546270, BioProject PRJNA692994 and BioProject PRJNA693004. Files for trees and heatmaps in Appendix A.

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
