# Peer review of "Taxonomic Re-Evaluation and Genomic Comparison of Novel Extracellular Electron Uptake-Capable Rhodovulum visakhapatnamense and Rhodovulum sulfidophilum Isolates"

_microorganisms, 2022, doi:10.3390/microorganisms10061235_

Round 1

Reviewer 1 Report

  • I'm glad to have the opportunity to review the manuscript. It raises an important topic: assessing taxonomic re-evaluation based on 16S, pufM phylogenetic analyses and re-classify of Rhodovulum visakhapatnamense.
  • The presented data are very interesting, but the manuscript needs improvement. In its current form, it is not sufficient to publish by Microorgansims.
  • I have made a few comments, aiming to improve the manuscript quality and readability.
  • Abstract need to rewrite, you need to speak in general not our group, our group. This article not report.
  • In abstract, you concentrate on AB26 isolate and no any results about AB19 isolate.
  • In key words, please add e Rhodovulum visakhapatnamense and Rhodovulum sulfidophilum. You can remove  environmental microbiology.
  • Line 28: please add Reference.
  • The introduction does not explain Rhodovulum visakhapatnamense and Rhodovulum sulfidophilum. You need to some information about 

    characteristics and taxonomy of Rhodovulum visakhapatnamense and Rhodovulum sulfidophilum.

  • There are many abbreviations used throughout the manuscript. A detailed list of abbreviations could be tabulated at the start of the manuscript. please review.
  • Font size, pleas unified in all text.
  • Lines60-63: This results not introduction.
  • Line 148: you add table about AB26 Genome Features and Expression Analysis, where is AB19 Genome Features and Expression Analysis?. Please add table about AB19.
  • The discussion sections also lack a more robust grounding of the proposed goal. Please expand.
  • In all text you used we many times espically in Conclusions, so please change to our study, our research, ........etc. Try to use othe expression.
  • Please add update 2022 References 

Reviewer 2 Report

This paper presents a huge amount of data on the genomics of extracellular electron uptake and related metabolisms of the purple nonsulfur bacteria Rhodovulum sulfidophilum and Rhodovulum visakhapatnamense. The authors conclude that two of the isolates they obtained from enrichment cultures and originally classified as strains of R. sulfidophilum are actually strains of R. visakhapatnamense, although from a phenotypic standpoint, these two species are nearly identical. The paper could be improved by considering some of the comments below and by the authors presenting actual conclusions concerning the various genes they describe rather than simply summarizing the results.

Specific comments:

1. On the title page, the authors names should be listed as given name first and family name second (they are currently listed just the opposite).

2. Abstract: I think this is the only time I caught this, but Rhodovulum species should be called "phototrophic" bacteria, not "photosynthetic" bacteria (that is, they eat light, they don't make it). The correct name was used thereafter as far as I could tell.

3. Use of the word "strain". Throughout the ms, the authors refer to "AB26" or "AB19" or "TIE" just as such. These strain designations should be preceded by the word "strain" (or "Strain" if beginning a sentence). This will remind the reader that these numbers are organisms and not genes or transcripts.

4. Methods: The methods in terms of growth conditions were very sketchy. a. For example, there are no methods described as to how the various organisms were grown--presumably phototrophically (anoxic/light)--but in what medium and at what temperature, light intensity, salinity, and etc. Also, none of the 4 metabolisms whose gene expression was evaluated in Fig. 4 are spelled out; how was growth on thiosulfate done (was this strictly photoautotrophic growth or were some organic compounds present?); how was growth on H2 done (how much H2, for example?); how was aerobic growth done (shaker, bubbling, on plates)?  b. The pufM sequencing method should be described or cited. c. It would be helpful to list Genbank numbers for the genome sequences in the Methods as well as at the end of the paper.

5. Results: Table 1. The value of 98.2% for protein coding genes in this genome sounds unusually, indeed almost unbelievably, high. Are the authors sure of this? Numbers this high are not often found in bacteria with genomes as large as these purple bacteria (~4.3 Mbp). Such values are more typical of species with tiny genomes. Line 161-the word "nonsulfur" (as in purple nonsulfur bacteria) is one word, not hyphenated. Lines 195-196. It would be better to say "...various inorganic compounds as electron donors...". As written (...various inorganic compounds for energy synthesis...) the phrase implies that these donors are oxidized in energy-conserving processes (i.e. proton translocation). This could be true during aerobic growth but not under phototrophic conditions where light is the energy source and no net electron acceptor (e.g. nitrate or sulfate) is being reduced.

6. Line 288: What is meant by the phrase "To compete within their niche"? Why would multi-drug transporters, for example, be necessary for these purple nonsulfur bacteria to compete in their natural habitats? Presumably these habitats are not loaded with antibiotics. This should be either re-phrased or deleted. Line 309: Is the nitrogenase present in strain AB26 a canonical Mo-containing nitrogenase? Line 319: Do you mean "ammonia assimilation" here where you say "nitrate assimilation"? Also, the legend to Fig. 7 mentions urea conversion (do you mean degradation?) and nitrogen fixation but neither of these are shown in the figure. Line 365: Alpha-, Beta-, and Gammaproteobacteria are taxonomic classes, not orders. Line 372 et seq: Can any additional conclusions be made as to what this hypothetical protein might participate in?

7. The conclusions section (Lines 413 et seq) gives very few conclusions. A nice summarizing figure showing how the genomic analyses presented in the paper have contributed to our understanding of pEEU would be very helpful. For example, a figure that showed the biochemical steps in pEEU with emphasis on newly discovered connections made from the genomic analyses performed in the paper would help to connect the data to the conclusions.

Round 2

Reviewer 1 Report

Accept in present form

Author Response

We thank you for accepting our manuscript in its present form!